# New Composite Contrast Agents Based on Ln and Graphene Matrix for Multi-Energy Computed Tomography

**DOI:** 10.3390/nano12234110

**Published:** 2022-11-22

**Authors:** Evgeniya V. Suslova, Alexei P. Kozlov, Denis A. Shashurin, Vladislav A. Rozhkov, Rostislav V. Sotenskii, Sergei V. Maximov, Serguei V. Savilov, Oleg S. Medvedev, Georgy A. Chelkov

**Affiliations:** 1Department of Chemistry, Lomonosov Moscow State University, 119991 Moscow, Russia; 2Faculty of Medicine, Lomonosov Moscow State University, 119991 Moscow, Russia; 3Joint Institute for Nuclear Research, 141980 Dubna, Russia; 4Laboratory of Experimental Pharmacology, Institute of Experimental Cardiology, National Medical Research Center of Cardiology Named after Academician E.I. Chazov, 121552 Moscow, Russia

**Keywords:** graphene nanoflakes, photon-counting computed tomography, multi-energy computed tomography, contrast agents (CAs), lanthanide, support, functionalization, X-ray, nanoparticle

## Abstract

The subject of the current research study is aimed at the development of novel types of contrast agents (CAs) for multi-energy computed tomography (CT) based on Ln–graphene composites, which include Ln (Ln = La, Nd, and Gd) nanoparticles with a size of 2–3 nm, acting as key contrasting elements, and graphene nanoflakes (GNFs) acting as the matrix. The synthesis and surface modifications of the GNFs and the properties of the new CAs are presented herein. The samples have had their characteristics determined using X-ray photoelectron spectroscopy, X-Ray diffraction, transmission electron microscopy, thermogravimetric analysis, and Raman spectroscopy. Multi-energy CT images of the La-, Nd-, and Gd-based CAs demonstrating their visualization and discriminative properties, as well as the possibility of a quantitative analysis, are presented.

## 1. Introduction

Computed tomography (CT) is an in vivo method for bioimaging [1,2] tissues and organs. Some types of tissues, such as those in the bones and the lungs, are easily visualized using roentgenologic equipment without the introduction of contrast agents (CAs). However, sometimes CT-based imaging studies require additional contrasting procedures. Traditional CAs include iodine (used to visualize blood vessels and mammary glands) and barium in the form of BaSO4 (used for studies of the digestive tract) [1,3], although, at present, the range of available CAs has been significantly expanded, with a particular focus on contrasts that allow for the visualization of hard tissues in the body [4]. There have also been developments and advancements in compounds with CA properties and among other therapeutic agents as well [2].

CT methods and types of equipment also continue to advance and develop, with a good example being the recent advances in semiconductor X-ray detectors, which allowed for the shift from traditional monochromatic CT towards multi-energy CT. This method not only allows for the evaluation of the total X-ray density of the studied tissues and organs but also the identification and visualization of up to six compounds that differ in the spectral lines (K-edges) of their constituent elements [5,6,7]. This approach is extremely important for biomedical research and clinical medicine because it allows for the performance of complex roentgenologic assessments including the visualization of tissues with similar X-ray density and, at the same time, the reduction in doses of radiation. Finally, this method also produces images with micrometer resolution [4].

It is possible to conduct multi-energy CT studies without additional contrasting, but the effectiveness of such a method can be significantly increased using CAs that include elements whose K-edges significantly differ from those of the common elements constituting biological tissues, i.e., the elements with a K-edge energy in the range of 50–90 keV. These include elements with Z = 64–83 [8,9,10]. Recently, CAs including Hf [4], Ta, Gd, Yb [8], W, Bi [11], Au [12], La, Lu, Ho [13], Eu, Er [14], and Dy [15] were studied. Among the lanthanides, the most frequently applied element is Gd [16].

In addition to CAs with a traditional molecular structure and, in particular, chelate-based CAs that reduce the toxicity of free Ln ions [17,18], CAs in the form of nanoparticles are being studied. Such CAs have certain advantages in terms of their effectiveness for diagnosis and therapy [3,11,19,20]. Specifically, the nanoparticles, depending on their size, have a tendency to stay in the body for longer periods compared to small molecules, which increases the CAs’ effectiveness and the possible duration of a roentgenologic study. Furthermore, nanoparticles can be modified to improve their affinity to specific biological molecules to ensure the targeted contrasting of relevant tissues. For example, Au nanoparticles may be functionalized with high-density lipoproteins [21] or bisphosphonates [22], while HfO_2_ nanoparticles can be modified with nitrilotriacetic acid to improve their affinity for calcium, which has made it possible to visualize microcracks in bones [4]. Nanoparticles also have some disadvantages, i.e., their tendency to aggregate, as well as their insolubility in water [23]. However, this can be prevented by modifying their surfaces with hydrophilic polymers, for example, with polyethylene glycol [24], or by using an inorganic matrix based on SiO_2_ [25,26,27] or carbon nanotubes [28].

The present study is aimed at developing a prototype of the novel class of CAs consisting of Ln_2_O_3_ nanoparticles supported by a graphene matrix. In addition, we have evaluated the possibility of the simultaneous detection and differentiation of CAs including using different Ln (La, Nd, and Gd) within single multi-energy CT scans to demonstrate the possibility of multi-contrasted imaging studies. A synthetic, universal approach allowing for the production of CAs with a wide range of Ln has thus been proposed.

## 2. Materials and Methods

### 2.1. Synthesis

Graphene nanoflakes (GNFs) were synthesized according to the method described in [29]. Hexane (99.8%, Reachim) was pyrolytically decomposed at 900 °C in the presence of a MgO template (S_BET_ = 140 m^2^ g^–1^) in a quartz tube reactor with a diameter of 50 mm under a nitrogen (99.999%, Logika Ltd., Moscow, Russia) flow of 1000 mL min^–1^ for 30 min. The MgO template was removed by boiling the samples in 10 wt. % HCl for 5 h, followed by washing them with distilled water. The obtained GNFs were dried at 80 °C for 24 h.

The oxidation of GNFs was performed in the presence of HNO_3_ (Chimmed, 70%, 1.4 g cm^−3^ density, and 99.999% purity) according to [30]. The GNFs were refluxed under HNO_3_ for 1 h, filtered, washed with H_2_O, and dried at 80 °C for 24 h. The oxidized GNFs are referred to further as GNFs_ox.

Contrast agents were prepared by impregnation of GNFs_ox with Ln(NO_3_)_3_·6H_2_O (Ln = La, Nd, and Gd) (99%, China Northern Rare Earth group High-Tech Co. Ltd., Baotou, China) in an ethanol solution with a target metal load of 10–30 wt. %. First, 0.30 g GNFs_ox were treated by sonification in 300 mL of ethanol to produce a relatively stable suspension. After that, an ethanol solution of Ln(NO_3_)_3_·6H_2_O was added. The solvent was evaporated at 60 °C within 4 h under continuing sonication. Subsequently, the powder samples were moved into a quartz tube reactor and heated to 400 °C at a rate of 3 °C∙min^−1^. The heating continued for 30 min. A study of the thermal annealing of the GNFs_ox at 400 °C was also carried out in order to obtain a correct comparison of the changes in the properties and chemical composition of GNFs_ox. The composite samples are referred to further as Ln/GNFs_ox (Ln = La, Nd, and Gd) and the annealed sample is referred to as GNFs_ox_40.

### 2.2. Physio-Chemical Analysis

Specific surface area was determined by applying the BET model to the low temperature nitrogen adsorption data, which were obtained using an Autosorb-1C/QMS (Quantachrome Inc., Boynton Beach, FL, USA) analyzer. Samples were degassed at 300 °C for 3 h prior to undergoing adsorption analysis.

High-resolution transmission electron microscopy (HRTEM) images were captured using a JEOL 2100 F/Cs (Jeol, Tokyo, Japan) microscope operated at 200 kV and equipped with UHR pole tip as well as a spherical aberration corrector (CEOS, Heidelberg, Germany) and EEL spectrometer (Gatan, Munich, Germany). Morphology and composition of the samples were characterized using a JEOL JSM-6390LA scanning electron microscope operating at 25 kV.

Thermogravimetric analysis (TG), differential thermogravimetric analysis (DTG), and differential scanning calorimetry (DSC) were performed using a Netzsch STA 449 PC LUXX thermal analyzer with a heating rate of 5 °C∙min^−1^ under an Ar (99.999%, Ltd. Logica, Moscow, Russia) atmosphere and connected to an Aoeolos quadruple mass-spectrometer.

X-ray diffraction patterns were recorded in the 2θ range of 10–80° using the Stadi-P (Stoe & Cie, Darmstadt, Germany) instrument equipped with a Cu K_α1_ radiation source.

The Raman spectra of carbon nanomaterials were registered using a LabRam HR800 UV (Horiba, Kyoto, Japan) spectrometer equipped with a 5 mW argon laser (514.5 nm). Each sample was analyzed by at least 3 points with a subsequent averaging of the results.

The content of oxygen and Ln—as well as its oxidation states—were confirmed by X-ray photoelectron spectroscopy (XPS) using the Kratos Axis Ultra DLD instrument with a monochromatic Al K_α_ source operated at hν = 1486.6 eV and 150 W (Shimadzu, Milton Keynes, UK). Survey XPS spectra were recorded with an analyzer pass energy of 160 eV and steps of 1 eV. High-resolution spectra were recorded with analyzer pass energy of 20 eV and steps of 0.05 eV.

### 2.3. Multi-Energy Computed Tomography

Multi-energy CT studies were performed using a custom multi-energy computed tomograph based on the WidePIX pixel detector set at single-photon-counting mode with direct conversion. The objects of the study were placed on a Standa 8MR190-2 [31] rotary platform. The X-ray source used was a SourceRay SB120-350 X-ray tube [32] operated at an anode voltage of 120 kVp with a current of 50 µA. Measurements were carried out with the X-ray tube preheated to 24.2 ± 0.4 °C. Circular projections of over 360° were acquired using a WidePIX detector with 15 Medipix 3RX (fine-pitch) chips using 1 mm thick Si sensor material [33]. The detector temperature was 19.75 ± 0.12 °C. Before the start of measurements, pixels were equalized for the detector through the noise level at a level of 8 keV. After that, the procedure for selecting idle and noisy pixels was carried out. The resulting set of projections for each energy threshold underwent a flat-field correction [34]. The projections obtained were used for 3D reconstruction using the Astra toolbox [35].

## 3. Results and Discussion

### 3.1. GNFs, GNFs_ox, and GNFs_ox_400 Characterizations

The TEM images of the GNFs demonstrated that they had a flat, polyhedral shape, thus replicating the template form (Figure 1a). The GNF particles contained 7–12 carbon layers in parallel with each other. The GNFs’ oxidation was necessary to produce sites for Ln stabilization and ensure their uniform surface distribution. After oxidation, the morphology of the GNFs changed: the edges of their particles became ragged (Figure 1b) because of the increase in the oxygen-containing terminal groups [30]. The nitrogen physisorption measurements determined the BET specific surface area (S_BET_) of GNFs_ox to be 374 m^2^∙g^−1^.

The Raman spectra of carbon materials typically contain two main lines. The D line at ~1360 cm^−1^ relates to the radial-breathing mode *A_lg_* of aromatic C6 rings in the graphene plane with conjugate *sp^2^*-carbon atoms. This mode can be resonantly excited under *π*-electrons’ optical transition in the corresponding graphene clusters. The G band at 1581 cm^−1^ corresponds to *E*_2*g*_ stretching vibrations of C6 rings in the graphene plane. All the samples had three 2D overtones at 2700 cm^−1^, (D + G) at 2960 cm^−1^, and 2G at ~3200 cm^−1^ in their Raman spectra due to the small size of the graphitic domains [36] and the curvature of the carbon layers [37]. At the same time, 2D, (D + G), and 2G overtones were observed in the Raman spectrum of graphene [38]. The defectiveness of the GNFs and GNFs_ox was estimated through the Raman spectra as the ratio of intensities of D and G lines I_D_/I_G_ (Figure 2). The ratios of I_D_/I_G_ of GNFs, GNFs_ox, and GNFs_ox_400 were 0.88, 0.90, and 0.86, which clearly indicated an increase in the defectiveness of the samples after their oxidation and a decrease in their defectiveness after the heat treatment.

The state and content of the oxygen and carbon were analyzed by XPS (Figure 3a). The oxygen content in the GNFs was equal to 0.7 at. % [30]. The total oxygen content in the GNFs_ox was 11.1 at. %. During the heat treatment of the GNFs_ox at 400 °C, portions of the functional groups were eliminated and the oxygen content decreased to 6.5 at. %.

The chemical states of the oxygen and carbon atoms were assessed using high-resolution O1s and C1s spectra (Figure 3b,c). According to the O1s spectra (Figure 3b), the sample GNFs_ox contained maximums at 531.5 and 533.2 eV corresponding to O=C and O–C; furthermore, after the heat treatment, new peaks appeared with energies of 530.6 and 532.6 eV. They corresponded to O- and C-OH states of oxygen [39]. This corresponds to the published data, indicating that the heat treatment of the GNFs_ox at a range of 300–500 °C leads to the degradation of the carboxyls and the keto groups on their surfaces, thus reflecting their transformations into less-oxidized hydroxyl and phenol groups [30,40].

The C1s spectra of GNFs_ox (Figure 3c) contained maximum levels at 284.6, 285.2, 286.4, 278.4, and 288.7 eV, corresponding to C-C (*sp^2^*), C-C (*sp^3^*) and the oxygen-containing groups C-O(H), C=O, and O=C-O, respectively. The content of C=O decreased after heat treatment, which was in line with the O1s spectra data.

### 3.2. Ln/GNFs_ox Characterizations

The standard methodology of GNFs’ impregnation by Ln assumes that the nitrates Ln(NO_3_)_3_^∙^6H_2_O used in the process must be completely thermally decomposed to Ln_2_O_3_ at 600 °C [41,42,43]. However, the goals of the present study required: (1) the preservation of the functional groups on the GNF_ox’s surface and (2) the protection of the GNF structure from excessive oxidation by the nitrogen oxides released during the thermal decomposition of nitrates. To ensure this, the thermal decomposition of Ln(NO_3_)_3_^∙^6H_2_O in the Ln/GNFs_ox samples was conducted under thermogravimetric (TG) control coupled with mass-spectrometry of waste gases under an Ar atmosphere. It was found that a temperature of 400 °C was optimal for nitrate decomposition without causing a dramatic loss of oxygen-containing surface groups (Figure 4).

The Ln content was assessed by EDX (Table 1). The contents of Ln according to EDX were higher than the experimental level (i.e., expected based on a ratio between the Ln precursor and CNFs used for synthesis) due to an inaccuracy in the method. For Nd and Gd, the variance was within 1 wt. %, which was in line with expectations. For La, however, the variance was higher. Perhaps the larger part of the carbon was burnt under the NO_x_ atmosphere resulting in additional weight loss (Figure 4a, Table 1).

The XRD patterns of the GNFs, GNFs_ox, and Ln/GNFs_ox are shown in Figure 5. The GNFs and GNFs_ox samples had three characteristic carbon peaks at 2θ = ~26, 44, and 65. However, after the Ln(NO_3_)_3_∙6H_2_O impregnation and heat treatment, the samples became amorphous, with only a very weak reflex at 2θ = ~41, which could be attributed to the Ln_3_O_4_NO_3_ phase [41].

The TEM images of the Ln/GNFs_ox (Figure 6) demonstrated that the Ln-containing particles were relatively uniform and possessed a size of 2–3 nm (Figure 6a–f). The particles were uniformly distributed over the support surface due to its functionalization with carboxyl groups, which corresponds to the conclusions from similar studies with metal/carbon composites [44]. The chosen mass load of Ln of 10 wt. % allowed for the prevention of the formation of agglomerates.

The chemical compositions of the surfaces of the samples were confirmed by XPS (Figure 7). The survey XPS spectra contained lines of oxygen, carbon, nitrogen, and Ln (Figure 7a). The La3d spectra of La/GNFs_ox (Figure 3c) contained high-intensity shake-up satellites with a binding energy of the La3d_5/2_ component equal to the 834.9 eV characterizing the La^3+^ in La_2_O_3_ or other compounds [45,46]. Similar patterns were observed for the samples with Nd and Gd: their Nd and Gd lines contained high-intensity shake-up satellites (Figure 7b) corresponding to the binding energies of the Nd4d and Nd3d_5/2_ components (982.6 and 122.7 eV) and Gd4d and Gd3d_5/2_ components (1187.2 and 142.8 eV). These spectra confirmed the presence of compounds of Nd and Gd with an oxidation state +3 [47,48,49].

The high-resolution spectra of the oxygen O1s (Figure 7c) and carbon C1s (Figure 7d) were decomposed according to ref. [39]. The O1s spectra had components with a binding energy of about 529 eV, which is characteristic of Ln_2_O_3_ (Figure 7c). However, intense components with binding energies of 530.7 and 531.4 eV, which could be attributed to oxygen in the form of hydroxides and carbonates, respectively, were also observed. Thus, the Ln in the samples was mainly associated with the hydroxyl and carboxyl groups. The presence of carbonates was also confirmed by the presence of components with a binding energy of 289.3 eV in the C1s spectra (Figure 7d). We believe that this was enough for the steady binding of the support (GNFs_ox) and metal particles. In addition, we expect that the Ln transformation into carbonates [50] led to a decrease in their chemical activity and an increase in the stability of nanoparticles, which is important for the intended application of the proposed composites as potential contrasting agents for biomedical research.

### 3.3. Multi-Energy Computed Tomography: A Phantom Study

In the present study, we synthesized the samples with La, Nd, and Gd as CAs in order to use elements with a similar chemical nature. This approach allows for the expansion of the synthetic process into one that is unified for all lanthanides and produces multiple sets of targeted CAs with minimal customizations of the synthetic approach itself. From a CT point of view, the choice of a set of Ln for a simultaneous determination should be such that the energy of the K-edge has a resolution of more than 5 keV. For the selected elements La, Nd, and Gd, these values are 38.9, 43.6, and 50.2 keV, respectively. The choice of Gd is consistent, because Gd is usually used as a CA for magnetic resonance imaging (MRI) [2] and meets the requirements of both MRI and CT. Lanthanum has an ordinal number Z = 57, which is less than the recommended minimum value Z = 64 [8]. It was included into the study due to the high volume of the data on its pharmacology, which should facilitate subsequent studies of the pharmacokinetics and pharmacodynamics of the novel CAs. Nd was chosen as an additional reference element and was used to demonstrate the possibility of the individual determination of lanthanides in multi-energy CT studies.

Initial multi-energy CT studies were performed using water solutions of the raw nitrates including Ln and other elements. The investigated samples packed in the Eppendorf tubes were examined by multi-energy CT using a polystyrene frame phantom (Appendix A). Ln was used in the form of nitrate water solutions. Additionally, samples of water, C_2_H_5_OH, Gd (Magnevist, Bayer Shering Pharma, AG, 75 mg∙mL^−1^), iodine solutions (Ultravist, Bayer Shering Pharma, AG, 370 mg∙mL^−1^), and bone were used.

The contrasting elements were identified using the custom photon energy-based criteria applied to the raw data from the slices produced from reconstructed 3D objects (the details of this analysis will be published separately). Three-dimensional reconstructions were produced for 12 individual energy thresholds, i.e., 64.5, 54.5, 49.7, 45.1, 40.6, 36.3, 32.1, 28, 24.1, 20.3, 16.6, and 13 keV (unlike in standard radiology works based on common FlatPanel detectors registering all photons under only one threshold). The lower threshold used to eliminate the noise was set as 8 keV. Considering that Si detectors have relatively low efficiency in registering high-energy photons (<10% for 60 keV), each energy threshold was assessed with the individual time of exposition needed to achieve 1000–1500 events per pixel for a flat field (without the study samples). The same exposition times were used for dark field and object projections because they are necessary for flat field correction. This approach allowed for the registering of only the photons with energy values exceeding the defined thresholds, which led to the increased scan contrast, and provided X-ray spectral data that could be translated into pseudo-colors that determine the elementary composition of the study objects after their 3D reconstruction. It is possible to determine which element is localized in each voxel because the absorption spectrum of each element is unique. After analyzing the behavior of the absorption coefficient in each voxel for each energy threshold, it is possible to determine the element that is characterized by a given behavior of the absorption coefficient.

In other words, the sequence of actions for obtaining data can be represented as follows:The creation of the X-ray projection of the study object at a certain angle for each energy threshold;Rotation of the study object to the next angle;Repetition of steps 1 and 2 until the object rotates 360 degrees;The creation of the flat field images for each energy threshold;The creation of the dark field images (to identify non-working and noisy detector pixels);Flat field correction for each energy threshold;3D reconstruction for each energy threshold;Analysis of the absorption coefficients for each voxel at each energy threshold;The calculation of the voxel K-line;The comparison of the voxel K-line with the absorption K-lines of known elements.

The analysis of the CT data was performed in the chosen regions of interest (ROI) corresponding to the locations of the study samples (Figure 8a). We built absorption attenuation curves for these ROIs (Figure 8b). The resulting spectra were assessed using the above-mentioned criterion determining the K-edges and, consequently, the contrasting element (i.e., I, La, Nd, Gd, etc.) present in the sample (Figure 8c). At the same time, the intensity of absorption (reflected in the reconstructed images through different brightness) demonstrated the concentration of the contrasting element (Figure 8d).

In addition to the initial test scans confirming the preliminary feasibility of the simultaneous determination and analysis of concentrations of different contrasting elements including Ln in multi-energy CT studies, calibration studies were performed. These studies were conducted using a series of water solutions of Ln(NO_3_)_3_∙6H_2_O (Ln = La, Nd, and Gd) with Ln concentrations equal to 0, 5, 10, 20, 40, and 80 mg∙mL^−1^. The range of the concentrations was chosen based on the results of preliminary imaging studies using a MARS BioImaging multi-energy CT device, which confirmed that La could be detected in water solutions with concentrations as low as 2.5 mg∙mL^−1^ [51]. The procedures described above were performed with these samples, and images of phantom slices were obtained (Figure 9a–c). After the application of the energy criterion, the determination of the contrasting elements (Figure 9d–f), and the assessment of their concentrations in the water solutions (Figure 9g–i), concentration/absorption curves were built. It was confirmed that all of the observed concentration correlations possessed a linear character (Figure 9j–l).

After the completion of the initial studies, which confirmed that Ln, Nd, and Gd could be concurrently identified and discriminated, the pilot scans of the composites Ln/GNFs_ox were then performed. These scans were carried out using the phantom with Ln/GNFs_ox samples mixed with powdered graphite to achieve an Ln concentration of 80 mg∙mL^−1^ and packed in Eppendorf tubes. The identification and discrimination of La, Nd, and Gd in these samples were conducted according to the algorithm described above. The CT image of the phantom is presented in Figure 10a and the color embodiment in Figure 10b. Furthermore, 3D videos are available in the additional materials section (Appendix A).

## 4. Conclusions

The present study consisted of four parts: (1) the synthesis and investigation of carbon nanoparticles, namely, graphene nanoflakes; (2) the synthesis and investigation of Ln-containing composites based on the graphene nanoflakes used as a carbon matrix; (3) the multi-energy CT assessment of the model water solutions of La, Nd, and Gd, as well as the demonstration that La, Nd, and Gd can be discriminated and quantitatively assessed; and (4) the multi-energy CT study of the composite CAs.

GNF nanoflakes were synthesized through hexane pyrolysis with the subsequent removal of the MgO template. The observed nanoflakes, as per TEM analysis, had the form of flat polyhedral particles with 7–12 graphene layers parallel to each other. After oxidation in the HNO_3_ solution, the surfaces of the GNFs were functionalized by carboxyl and hydroxyl groups. The total oxygen content according to XPS was equal to 11.1 at. %. This step was necessary to achieve the stronger interaction of the Ln(NO_3_)_3_∙6H_2_O precursor with the carboxyl groups that additionally coordinate La ions. The high BET surface area (374 m^2^∙g^−1^) and the presence of the functional groups led to uniform surface impregnation by the nitrates of Ln (Ln = La, Nd, and Gd). The heat decomposition of Ln(NO_3_)_3_∙6H_2_O in Ln/GNFs_ox resulted in the formation of Ln-containing particles with a size of 2–3 nm according to TEM. According to XPS, the Ln was in the Ln^3+^ ionic state and chemically combined with the GNF surface. The formation of Ln carbonates was characterized by low solubility, and chemical activity was also observed.

The multi-energy CT assessment of the proposed composites was performed in two steps. First, the model water solutions of Ln in the form of nitrates Ln(NO_3_)_3_∙6H_2_O (Ln = La, Nd, and Gd) were assessed to validate the feasibility of the simultaneous detection and discrimination of these elements as well as the assessment of their concentrations in the study objects. Second, test scans of the composite CAs were performed. It was confirmed that the proposed Ln/GNFs_ox composites can be detected and determined using multi-energy CT.

The results presented above have confirmed that the proposed Ln/GNFs_ox composites have the general properties required for their potential implementation as contrasting agents in biomedical research. Nevertheless, their application requires the further assessment of their stability in biological media as well as detailed studies of their pharmacological and toxicological properties. Specifically, it should be confirmed that no dissociation of the metal nanoparticles from the GNF matrix or formation of water-soluble Ln compounds occurs, meaning that the contrasting elements remain biochemically inactive, and their distribution remains to be further driven by the characteristics of the matrix. This might require the further optimization of the composite structure (i.e., the application of core–shell structures to ensure the full chemical inactivation of Ln nanoparticles or the adjustment of GNF’s surface characteristics to ensure the optimal distribution of the composites in the biologic medias) until the desired pharmacological properties are achieved. This work is currently ongoing and will be presented in further publications.

## Figures and Tables

**Figure 1 nanomaterials-12-04110-f001:**
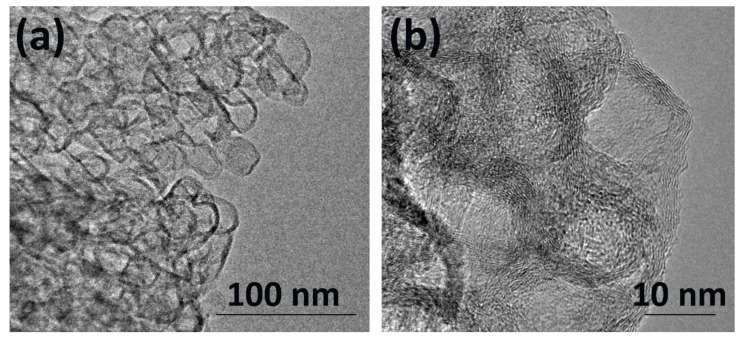
TEM images of GNFs (**a**) and oxidized GNFs_ox (**b**).

**Figure 2 nanomaterials-12-04110-f002:**
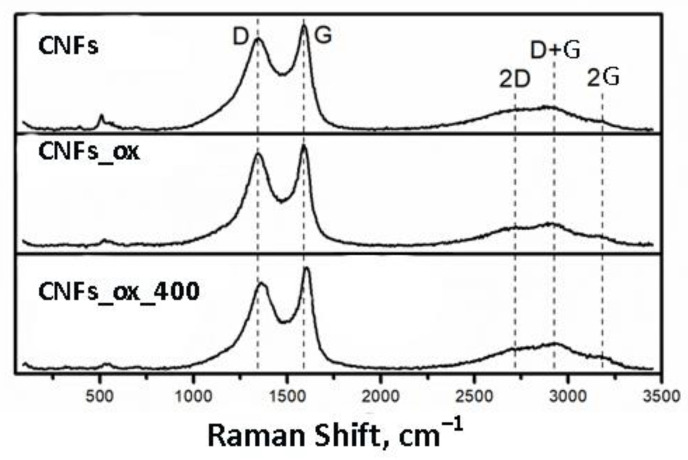
The Raman spectra of GNFs, GNFs_ox, and GNFs_ox_400.

**Figure 3 nanomaterials-12-04110-f003:**
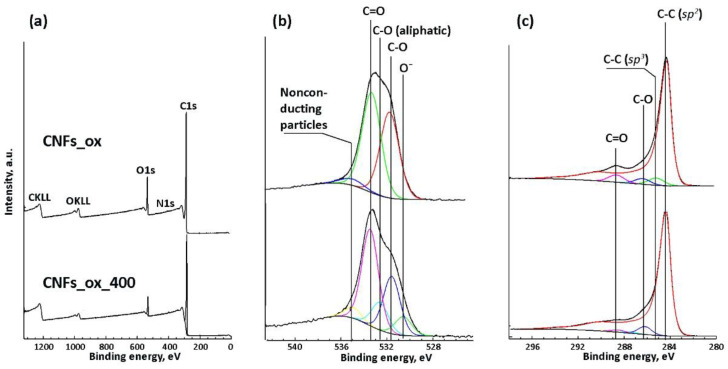
Survey (**a**), O1s (**b**), and C1s (**c**) XPS spectra of GNFs_ox (top) and GNFs_ox_400 (bottom).

**Figure 4 nanomaterials-12-04110-f004:**
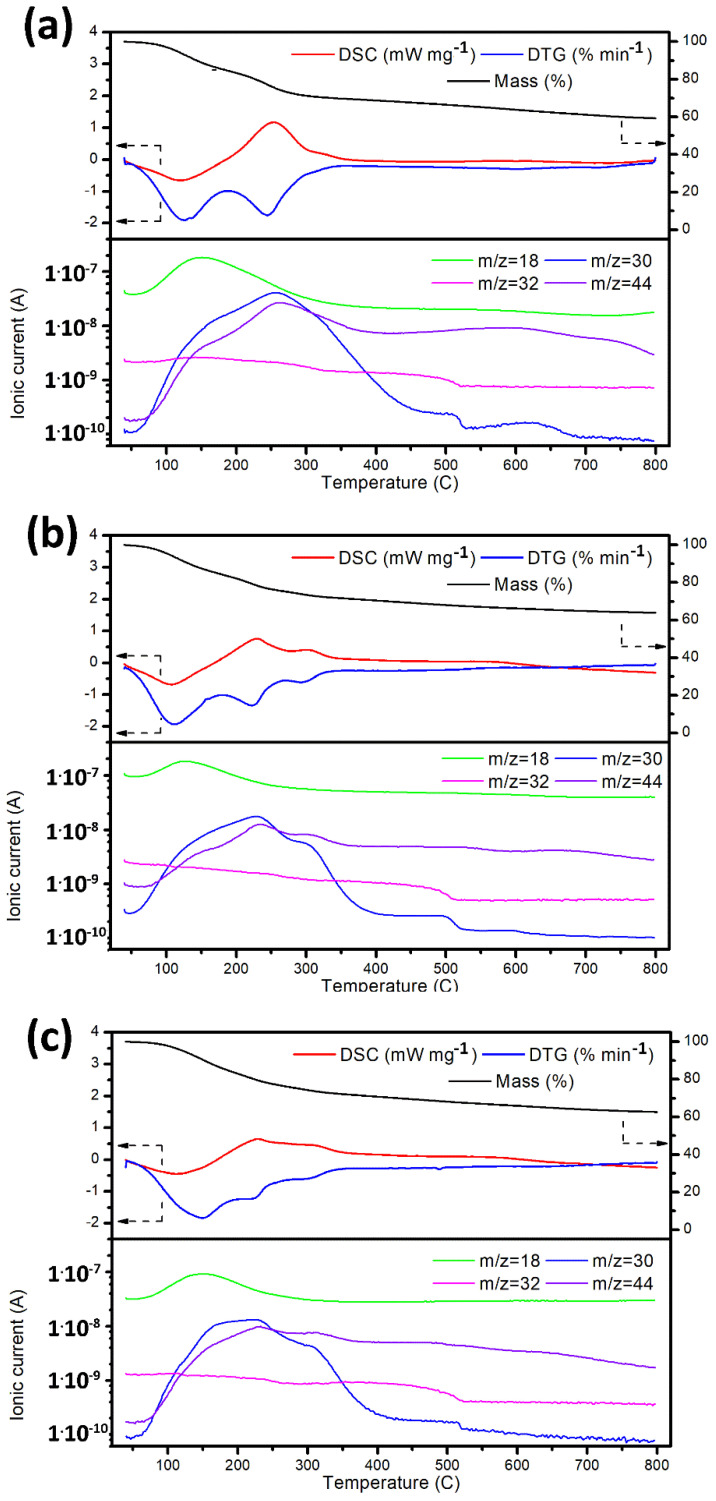
TG, DTG, and DSC curves (top) and gas products (bottom) of La/GNFs_ox (**a**), Nd/GNFs_ox (**b**), and Gd/GNFs_ox (**c**) heat treatments under Ar atmosphere.

**Figure 5 nanomaterials-12-04110-f005:**
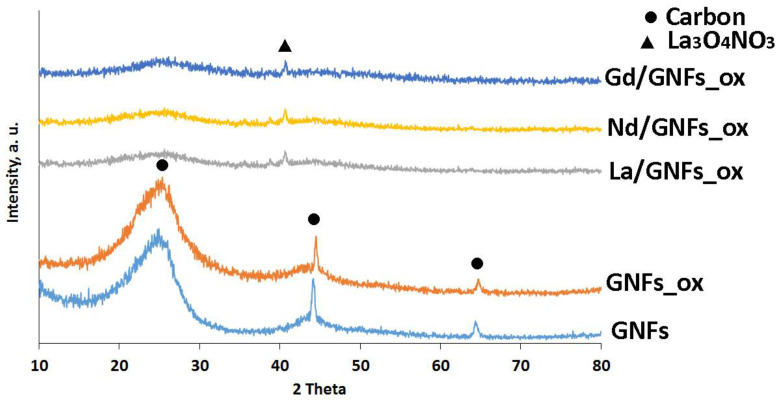
X-ray data of GNFs, GNFs_ox, and Ln/CNFs_ox.

**Figure 6 nanomaterials-12-04110-f006:**
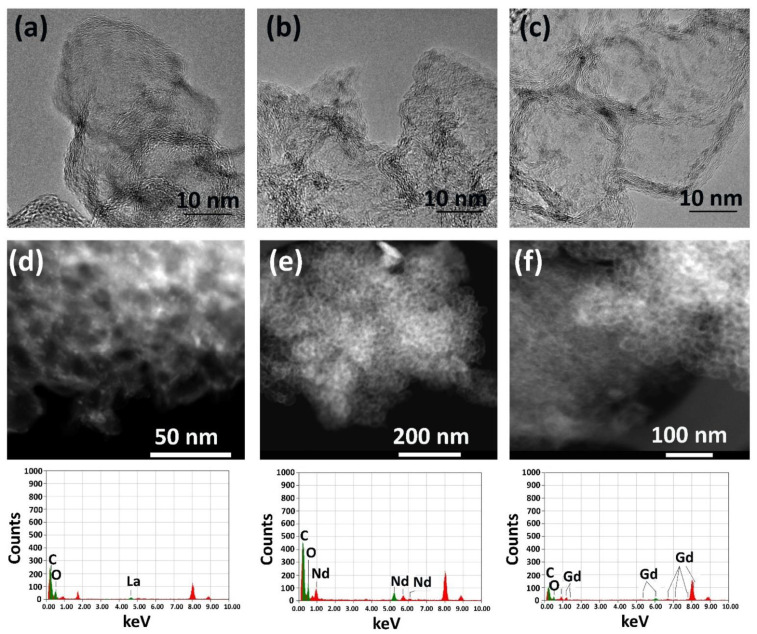
HRTEM (top row), HAADF-STEM (middle row) images and EELS spectra (lower row) of Ln/CNFs_ox particles: La/CNFs_ox (**a**,**d**), Nd/CNFs_ox (**b**,**e**), and Gd/CNFs_ox (**c**,**f**).

**Figure 7 nanomaterials-12-04110-f007:**
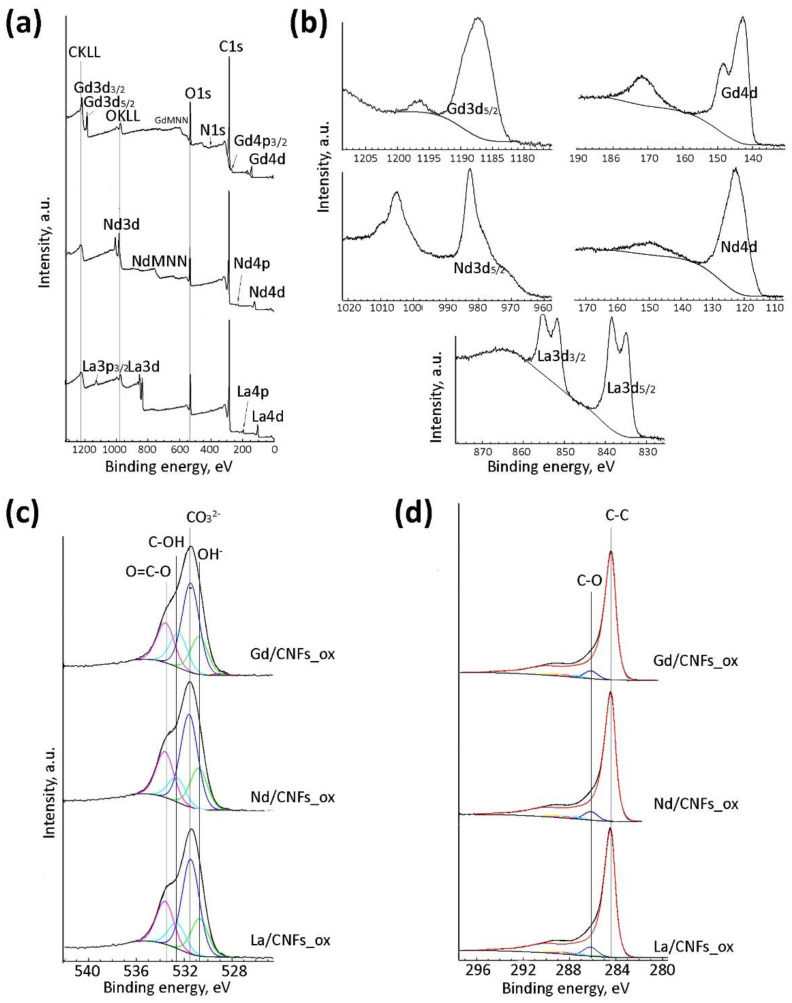
Survey XPS (**a**); La3d, Nd3d, Gd3d (**b**); O1s (**c**) and C1s (**d**) spectra of samples Ln/CNFs_ox.

**Figure 8 nanomaterials-12-04110-f008:**
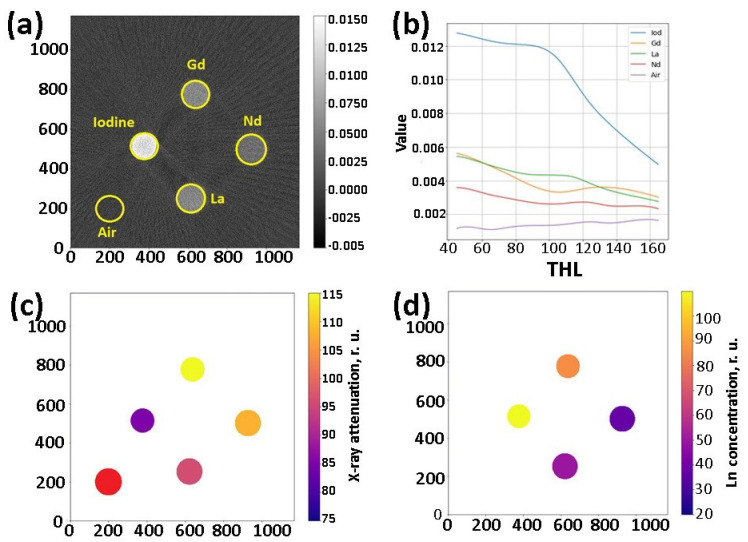
(**a**) CT image of the phantom at 45 thl, (**b**) dependence of gamma radiation absorption on the energy (detector threshold) in the ROI, (**c**) application of the criterion, and (**d**) evaluation of the molar concentration of the CA. Water solutions with the following concentrations of the contrasting elements were used: iodine—140 mg·mL^−1^, Gd—74 mg·mL^−1^, La—40 mg·mL^−1^, and Nd—40 mg·mL^−1^.

**Figure 9 nanomaterials-12-04110-f009:**
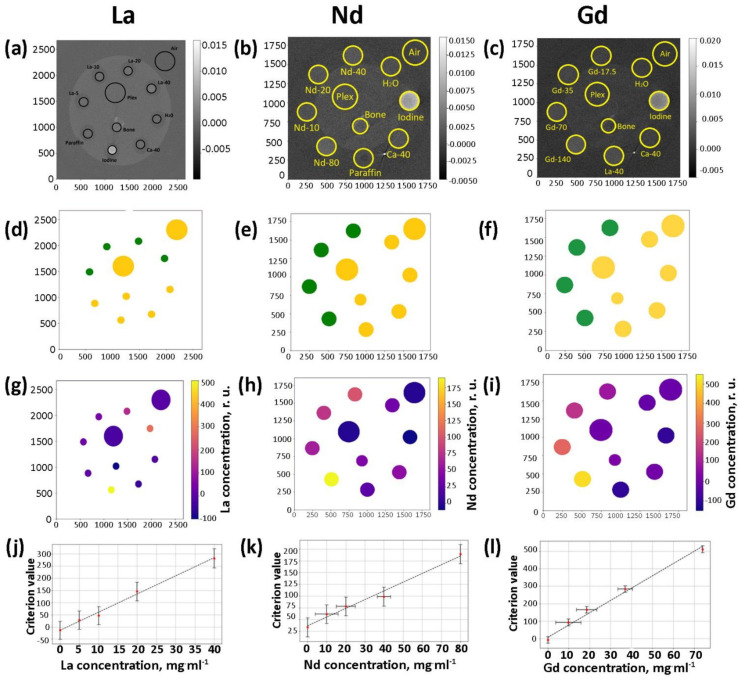
Upper row: CT images of phantom with the standard samples (water, iodine, paraffin, bone, and plex) and the investigated water solutions of La(NO_3_)_3_·6H_2_O (**a**), Nd(NO_3_)_3_·6H_2_O (**b**), and Gd(NO_3_)_3_·6H_2_O (**c**). The numbers shown in the images reflect the concentration of the contrasting element in the study solution. Second row: results of application of the photon energy criteria. Green corresponds to La (**d**), Nd (**e**), and Gd (**f**), while yellow corresponds to samples with no contrasting elements detected by criteria. Third row: the estimation of molar concentrations of La (**g**), Nd (**h**), and Gd (**i**). Bottom row: dependence of simulated concentration criteria from experimental concentrations of water solutions of La(NO_3_)_3_·6H_2_O (**j**), Nd(NO_3_)_3_·6H_2_O (**k**), and Gd(NO_3_)_3_·6H_2_O (**l**).

**Figure 10 nanomaterials-12-04110-f010:**
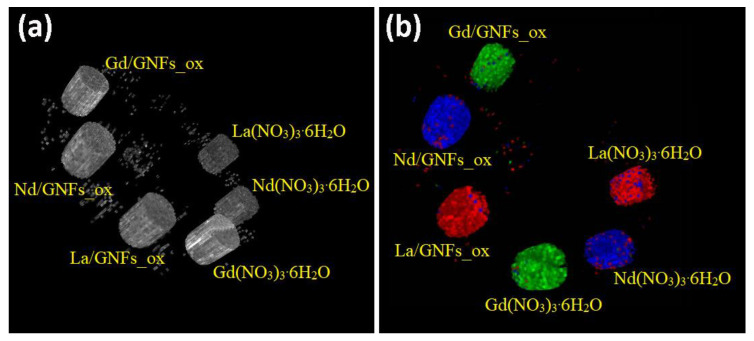
The 3D CT images of phantom with Ln/GNFs_ox (**a**) and color division of La/GNFs_ox (red), Nd/GNFs_ox (blue), and Gd/GNFs_ox (green) according to criteria (**b**).

**Table 1 nanomaterials-12-04110-t001:** The weight content of Ln in the Ln/CNFs_ox samples.

Ln/GNFs_ox	Ln Content, wt. %		Weight Residue at 400 °C under Ar Atmosphere, wt. %
	Experimental	EDX	TG
La	9.3	12.9	68.7
Nd	9.6	9.7	70.4
Gd	10.8	11.6	70.8

## Data Availability

Not applicable.

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
