# Peer review of "New Composite Contrast Agents Based on Ln and Graphene Matrix for Multi-Energy Computed Tomography"

_nanomaterials, 2022, doi:10.3390/nano12234110_

Round 1

Reviewer 1 Report

1. The authors have demonstrated the possibility of simultaneous detection and differentiation of the CAs including different Ln (La, Nd, Gd) within single multi-energy CT scans to demonstrate the possibility of multi-contrasted imaging studies.

2. There are typos and grammatical issues that warrant further revisions. Name a few examples, However, in many cases, CT-based imaging studies require additional contracting (contrasting); CT methods and equipment (types of equipment) also continue to develop.; The observed (obtained) GNFs were dried at80 оС for 24 h.; The present study was aimed (aimed) on developing the (a) novel type of CAs consisting of Ln2O3 nanoparticles supported by graphene matrix.

3. Line 194, The Ln content was assessed by EDX (table 1). The content of only La was more than the experimental. This does not meet the Table 1 content.

4. More issues need advanced discussion, ideally in a new section 3.4, to make current results practical. For example, the rationale for the preferred selection of GNFs_ox over GNFs and GNFs_ox_400 is not clear. The concentrations used for various tested CAs are different. The safety, efficacy, selectivity toward biochemical targets, and specificity of the proposed contrast agent compared to other CAs, for example, the widely used Magnevist.

Author Response

Reviewer#1

  1. There are typos and grammatical issues that warrant further revisions. Name a few examples, However, in many cases, CT-based imaging studies require additional contracting (contrasting)ï¼›CT methods and equipment (types of equipment) also continue to develop.ï¼› The observed (obtained) GNFs were dried at 80 оС for 24 h.ï¼› The present study was aimed (aimed) on developing the (a) novel type of CAs consisting of Ln2O3 nanoparticles supported by graphene matrix.

Answer:

We performed further proof-reading of the revised manuscript. Additionally, the text underwent a proof-reading by external native English speaker. All typos and grammatical issues were corrected.

  1. Line 194, The Ln content was assessed by EDX (table 1). The content of only La was more than the experimental. This does not meet the Table 1 content.

Answer:

The contents of Ln according to EDX were higher than the experimental (i.e. expected basing on ratio between Ln precursor and CNFs used for synthesis) due to a method inaccuracy. For Nd and Gd the variance was within 1 wt. % that was in line with expectations. For La the variance, however, was higher. Perhaps, the larger part of carbon was burnt under NOx atmosphere resulting in additional weigh loss.  Necessary data (weight residue at 400 oC under Ar atmosphere) were added to the table 1 and the explanatory text was updated to include the above.

  1. More issues need advanced discussion, ideally in a new section 3.4, to make current results practical. For example, the rationale for the preferred selection of GNFs_ox over GNFs and GNFs_ox_400 is not clear.

Answer:

The general approach to obtaining metal-deposited carbon composites is based on an impregnation of the carbon matrix with a metal-containing precursor followed by annealing. For a more uniform distribution of metal-containing particles on the carbon surface it can be functionalized with carboxyl, hydroxyl or similar groups that can be achieved through its oxidation. This approach was used in the present work and was the reason why the oxidized GNFs rather than raw GNFs were chosen for further synthesis of the composites. Also, we expect that the oxidized GNFs will produce more stable suspension and will demonstrate lower trend towards aggregation in biologic media such as blood serum. The samples GNF_ox_400 (oxidized GNFs, i.e. GNF_ox, that underwent further heat treatment) were produced to evaluate impact of the further synthetic processes towards GNF_ox structure. The synthetic approach used in our work included additional heat treatment of the impregnated GNFs to achieve decomposition of Ln nitrates and formation of insoluble carbonates. So we synthetized additional experimental GNF_ox_400 samples to assess the properties of the carbon matrix after additional treatment and to confirm that it doesn’t lead to significant loss of functionality. These samples were not intended for further synthetic use. 

The concentrations used for various tested CAs are different.

Answer:

The range of the concentrations (0-80 mg/ml) was chosen basing on results of preliminary imaging studies using MARS BioImaging multi-energy CT device confirming that La could be detected in water solutions with concentrations down to 2.5 mg/ml. We added a note regarding this into respective section of Results. The concentrations used in the imaging studies with simultaneous detection and discrimination of the different agents were chosen experimentally to achieve comparable X-ray density of the samples for their better visualization in 2D slices.

The safety, efficacy, selectivity toward biochemical targets, and specificity of the proposed contrast agent compared to other CAs, for example, the widely used Magnevist.

Answer:

The current manuscript summarizes results of the first “proof-of-concept” part of the larger research project aimed on development of non-specific and targeted contrasting agents for multi-energy CT based on composites with inorganic matrix and Ln nanoparticles as contrasting elements. In this study we demonstrated the possibility of the synthesis of the composites with desired structure, assessed their physico-chemical properties and also demonstrated the possibility of their simultaneous visualization, determination and quantitative analysis. We also obtained some preliminary data allowing to assume that the composites will be stable and relatively inactive in the biologic environment (i.e. confirmed that Ln nanoparticles form stable covalent bonds with the carbon matrix and that Ln precursors are transformed into insoluble and chemically inactive carbonates).

As next step in our research project we plan conducting comprehensive studies of the pharmacological and toxicological properties of the proposed composites. These studies may also lead to further optimization of their structure to increase their stability and/or further chemical modification of the carbon matrix surface to tailor their pharmacokinetics. As a result of this work we plan to produce not only effective and safe non-specific contrasts but also sets of the targeted contrasting agents with different affinity to the various biochemical targets to allow complex multi-contrasting imaging studies.

We feel that at this stage these considerations are too general to include them into Results and Discussion of the proposed paper. However we updated the Conclusions of the current paper to clarify the boundaries of the current study as well as our next steps in development of the proposed composites.

Reviewer#2

The work reported by Suslova et al. is likely to deserve interest in the field of the search on new contrasting materials for diagnostic imaging in vivo and the opportunity to have multi-contrasting systems is always attractive.

Nevertheless, in my opinion the description of experimental procedures and the presentation of results is not clear enough for a non-expert readerIn particular, the description of the CT experiments should be improved.

Answer:

We re-worked description of the multi-energy CT methodology and descriptions of the individual imaging studies to make them clearer.

How is composed the phantom of figure 8?

Answer:  Original fig.8 was intended to show general shape of the phantom used in our imaging studies, with details of the samples used in each individual imaging study shown later in respective figures and their descriptions. We recognize that this could be misguiding so the fig. 8 was moved to supplementary materials to avoid confusion. Additionally, we re-worked the descriptions of individual studies and the figures with their results to ensure that they clearly outline each sample assessed there.

A specific description of what exaclty is contained in the different eppendorf tubes would help in the understanding of the images reported in figure 11. 

Answer:

These scans were carried out using the phantom with Ln/GNFs_ox samples mixed with powder graphite to achieve Ln concentration of 80 mg.ml-1. We added identifiers of the samples to the figure itself (fig. 10 in revised manuscript) and included a note about composition of these samples into description of this imaging experiment.

Moreover, for what concerns the possible application of the system described for in vivo imaging, its stability toward dissociation of coordinated Ln(III) ions in a biological environment (i.e. serum) should be provided. Dissociation of free Ln(III) ions would be responsible of big toxicity  issues once administered to patients. 

Answer:

The proposed composites do not contain free Ln(III) ions. All Ln(III) is bound into stable and non-soluble form of Ln carbonates that was confirmed by XPS data. So we can expect that no toxicity related to dissociation of free Ln(III) ions will be observed.

With that said, we fully concur that the proposed composites require further assessment specifically from perspective of their use in biological objects. The current manuscript summarizes results of the first “proof-of-concept” part of the larger research project where we synthetized the composites with desired structure, assessed their physico-chemical properties and demonstrated the possibility of their simultaneous visualization, determination and quantitative analysis using multi-energy CT. As next step in our research project we plan conducting comprehensive studies of the pharmacological and toxicological properties of the proposed composites. These studies may also lead to further optimization of their structure to increase their stability and/or further chemical modification of the carbon matrix surface to tailor their pharmacokinetics. As a result of this work we plan to produce not only effective and safe non-specific contrasts but also sets of the targeted contrasting agents with different affinity to the various biochemical targets to allow complex multi-contrasting imaging studies.

We updated the conclusions of the current paper to clarify the boundaries of the current study as well as our next steps in development of the proposed composites.

Reviewer 2 Report

The work reported by Suslova et al. is likely to deserve interest in the field of the search on new contrasting materials for diagnostic imaging in vivo and the opportunity to have multi-contrasting systems is always attractive. Nevertheless, in my opinion the description of experimental procedures and the presentation of results is not clear enough for a non-expert reader.  In particular, the description of the CT experiments should be improved. How is composed the phantom of figure 8? a specific description of what exaclty is contained in the different eppendorf tubes would help in the understanding of the images reported in figure 11. 

Moreover, for what concerns the possible application of the system described for in vivo imaging, its stability toward dissociation of coordinated Ln(III) ions in a biological environment (i.e. serum) should be provided. Dissociation of free Ln(III) ions would be responsible of big toxicity  issues once administered to patients. 

Author Response

Reviewer#2

The work reported by Suslova et al. is likely to deserve interest in the field of the search on new contrasting materials for diagnostic imaging in vivo and the opportunity to have multi-contrasting systems is always attractive.

Nevertheless, in my opinion the description of experimental procedures and the presentation of results is not clear enough for a non-expert readerIn particular, the description of the CT experiments should be improved.

Answer:

We re-worked description of the multi-energy CT methodology and descriptions of the individual imaging studies to make them clearer.

How is composed the phantom of figure 8?

Answer:  

Original fig.8 was intended to show general shape of the phantom used in our imaging studies, with details of the samples used in each individual imaging study shown later in respective figures and their descriptions. We recognize that this could be misguiding so the fig. 8 was moved to supplementary materials to avoid confusion. Additionally, we re-worked the descriptions of individual studies and the figures with their results to ensure that they clearly outline each sample assessed there.

A specific description of what exaclty is contained in the different eppendorf tubes would help in the understanding of the images reported in figure 11. 

Answer:

These scans were carried out using the phantom with Ln/GNFs_ox samples mixed with powder graphite to achieve Ln concentration of 80 mg.ml-1. We added identifiers of the samples to the figure itself (fig. 10 in revised manuscript) and included a note about composition of these samples into description of this imaging experiment.

Moreover, for what concerns the possible application of the system described for in vivo imaging, its stability toward dissociation of coordinated Ln(III) ions in a biological environment (i.e. serum) should be provided. Dissociation of free Ln(III) ions would be responsible of big toxicity  issues once administered to patients. 

Answer:

The proposed composites do not contain free Ln(III) ions. All Ln(III) is bound into stable and non-soluble form of Ln carbonates that was confirmed by XPS data. So we can expect that no toxicity related to dissociation of free Ln(III) ions will be observed.

With that said, we fully concur that the proposed composites require further assessment specifically from perspective of their use in biological objects. The current manuscript summarizes results of the first “proof-of-concept” part of the larger research project where we synthetized the composites with desired structure, assessed their physico-chemical properties and demonstrated the possibility of their simultaneous visualization, determination and quantitative analysis using multi-energy CT. As next step in our research project we plan conducting comprehensive studies of the pharmacological and toxicological properties of the proposed composites. These studies may also lead to further optimization of their structure to increase their stability and/or further chemical modification of the carbon matrix surface to tailor their pharmacokinetics. As a result of this work we plan to produce not only effective and safe non-specific contrasts but also sets of the targeted contrasting agents with different affinity to the various biochemical targets to allow complex multi-contrasting imaging studies.

We updated the conclusions of the current paper to clarify the boundaries of the current study as well as our next steps in development of the proposed composites.

Round 2

Reviewer 2 Report

The answers from authors and the revisions included in the new version of the manuscript satisfy my previous observations, thus I think the manuscript has been sufficiently improved to warrant publication in Nanomaterials.